# Researching COVID to Enhance Recovery (RECOVER) adult study protocol: Rationale, objectives, and design

Leora I. Horwitz[1]*, Tanayott Thaweethai[2], Shari B. Brosnahan[3], Mine S. Cicek[4], Megan L. Fitzgerald[5], Jason D. Goldman[6], Rachel Hess[7], S. L. Hodder[8], Vanessa L. Jacoby[9], Michael R. Jordan[10], Jerry A. Krishnan[11], Adeyinka O. Laiyemo[12], Torri D. Metz[13], Lauren Nichols[14], Rachel E. Patzer[15], Anisha Sekar[5], Nora G. Singer[16], Lauren E. Stiles[17], Barbara S. Taylor[18], Shifa Ahmed[2], Heather A. Algren[19], Khamal Anglin[20], Lisa Aponte-Soto[21], Hassan Ashktorab[12], Ingrid V. Bassett[22], Brahmchetna Bedi[23], Nahid Bhadelia[24], Christian Bime[25], Marie-Abele C. Bind[2], Lora J. Black[26], Andra L. Blomkalns[27], Hassan Brim[28], Mario Castro[29], James Chan[2], Alexander W. Charney[30], Benjamin K. Chen[31], Li Qing Chen[32], Peter Chen[33], David Chestek[34], Lori B. Chibnik[35], Dominic C. Chow[36], Helen Y. Chu[37], Rebecca G. Clifton[38], Shelby Collins[23], Maged M. Costantine[39], Sushma K. Cribbs[23], Steven G. Deeks[40], John D. Dickinson[41], Sarah E. Donohue[42], Matthew S. Durstenfeld[43], Ivette F. Emery[44], Kristine M. Erlandson[45], Julio C. Facelli[46], Rachael Farah-Abraham[47], Aloke V. Finn[48], Melinda S. Fischer[18], Valerie J. Flaherman[49], Judes Fleurimont[50], Vivian Fonseca[51], Emily J. Gallagher[52], Jennifer C. Gander[53], Maria Laura Gennaro[54], Kelly S. Gibson[55], Minjoung Go[56], Steven N. Goodman[57], Joey P. Granger[58], Frank L. Greenway[59], John W. Hafner[60], Jenny E. Han[61], Michelle S. Harkins[62], Kristine S. P. Hauser[63], James R. Heath[64], Carla R. Hernandez[65], On Ho[66], Matthew K. Hoffman[67], Susan E. Hoover[26], Carol R. Horowitz[68], Harvey Hsu[69], Priscilla Y. Hsue[40], Brenna L. Hughes[70], Prasanna Jagannathan[56], Judith A. James[71], Janice John[72], Sarah Jolley[73], S. E. Judd[74], Joy J. Juskowich[75], Diane G. Kanjilal[76], Elizabeth W. Karlson[77], Stuart D. Katz[78], J. Daniel Kelly[40], Sara W. Kelly[79], Arthur Y. Kim[76], John P. Kirwan[80], Kenneth S. Knox[69], Andre Kumar[56], Michelle F. Lamendola-Essel[78], Margaret Lanca[81], Joyce K. Lee-Iannotti[82], R. Craig Lefebvre[83], Bruce D. Levy[84], Janet Y. Lin[34], Brian P. Logarbo, Jr.[85], Jennifer K. Logue[86], Michele T. Longo[87], Carlos A. Luciano[88], Karen Lutrick[89], Shahdi K. Malakooti[90], Gail Mallett[91], Gabrielle Maranga[1], Jai G. Marathe[92], Vincent C. Marconi[93], Gailen D. Marshall[94], Christopher F. Martin[47], Jeffrey N. Martin[95], Heidi T. May[96], Grace A. McComsey[97], Dylan McDonald[37], Hector Mendez-Figueroa[98], Lucio Miele[99], Murray A. Mittleman[100], Sindhu Mohandas[101], Christian Mouchati[90], Janet M. Mullington[102], Girish N. Nadkarni[103], Erica R. Nahin[78], Robert B. Neuman[104], Lisa T. Newman[105], Amber Nguyen[2], Janko Z. Nikolich[106], Igho Ofotokun[47], Princess U. Ogbogu[107], Anna Palatnik[108], Kristy T. S. Palomares[109], Tanyalak Parimon[33], Samuel Parry[110], Sairam Parthasarathy[25], Thomas F. Patterson[111], Ann Pearman[90], Michael J. Peluso[112], Priscilla Pemu[113], Christian M. Pettker[114], Beth A. Plunkett[115], Kristen Pogreba-Brown[116], Athena Poppas[117], J. Zachary Porterfield[118], John G. Quigley[119], Davin K. Quinn[120], Hengameh Raissy[121], Candida J. Rebello[122], Uma M. Reddy[123], Rebecca Reece[75], Harrison T. Reeder[2], Franz P. Rischard[124], Johana M. Rosas[78], Clifford J. Rosen[44], Nadine G. Rouphael[23], Dwight J. Rouse[125], Adam M. Ruff[29], Christina Saint Jean[1], Grecio J. Sandoval[38], Jorge L. Santana[126], Shannon M. Schlater[127], Frank C. Sciurba[128], Caitlin Selvaggi[2], Sudha Seshadri[129], Howard D. Sesso[130], Dimpy P. Shah[131], Eyal Shemesh[132], Zaki A. Sherif[133], Daniel J. Shinnick[2], Hyagriv N. Simhan[134], Upinder Singh[135], Amber Sowles[13], Vignesh Subbian[136], Jun Sun[11], Mehul S. Suthar[137], Larissa J. Teunis[138], John M. Thorp, Jr.[139], Amberly Ticotsky[72], Alan T. N. Tita[140], Robin Tragus[141], Katherine R. Tuttle[142], Alfredo E. Urdaneta[27], P. J. Utz[135], Timothy M. VanWagoner[143], Andrew Vasey[41], Suzanne D. Vernon[144], Crystal Vidal[1], Tiffany Walker[23], Honorine D. Ward[145], David E. Warren[146], Ryan M. Weeks[147], Steven J. Weiner[38], Jordan C. Weyer[148], Jennifer

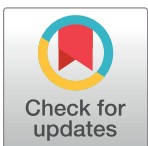

**Data Availability Statement:** No datasets were generated or analysed during the current study. All

relevant data from this study will be made available upon study completion.

**Funding:** National Institutes of Health (NIH) Other Transactional Authority Agreements OT2HL161847 (SDK, LIH), OT2HL161841 (ASF), OT2HL156812 (LTN). https://www.nih.gov/ The funder did have input into study design but did not and will not have a role in data collection and analysis, decision to publish, or preparation of the manuscript.

**Competing interests:** I have read the journal's policy and the authors of this manuscript have the following competing interests: Helen Chu reported consulting for Merck, GSK, Pfizer, Ellume, Janssen, Vindico CME, and the Bill and Melinda Gates Foundation, and receiving research support from Gates Ventures, Ellume, and Sanofi Pasteur. She also serves as a co-investigator on studies funded by Pfizer, Novavax, and GSK. Maged Costantine reported receiving grant support for work not related to RECOVER work/publications from Baxter International and Siemens Healthcare and personal consulting fees not related to this paper from Progenity, Quidel Ortho, and Siemens Healthcare. Kristine Erlandson reported research funding from Gilead Sciences and consulting payments from Gilead Sciences, Merck, and ViiV Pharmaceuticals, all paid to the University of Colorado. Emily Gallagher reported consulting for Novartis, Flare Therapeutics and Seagen. Edward Gardner reported research support (clinical trials) from Gilead Sciences, ViiV Healthcare, and Cepheid. Jason Goldman reported research support from Gilead, Eli Lilly and Regeneron; grants from Gilead, Merck (BARDA); personal fees for consulting from Gilead, Eli Lilly; and non-financial support from Adaptive Biotechnologies and Labcorp/Monogram Biosciences outside the submitted work. Timothy Heinrich reported grant support from Merck Inc. and consulting fees from Roche. Rachel Hess reported serving as Data Safety Monitoring Board member for Astellas Pharmaceuticals unrelated to the current work. Leora Horwitz reported being a member of the National Academy of Medicine Committee on the Long-Term Health Effects Stemming from COVID-19 and Implications for the Social Security Administration. Priscilla Hsue reported receiving honoraria from Gilead and Merck unrelated to study topic, receiving study drug from Regeneron unrelated to study topic, and receiving a research grant from Novartis. Judith James reported OMRF has licensed her IP to Progentec Biosciences, has received grant support from Progentec Biosciences, and serves on Advisory Committees to Glaxo Smith Klein, Merck and Novartis. Arthur Kim reports providing educational materials to Clinical Care Options and

**L. Wheeler[148], Sidney W. Whiteheart[149], Zanthia Wiley[47], Natasha J. Williams[150], Juan P. Wisnivesky[103], John C. Wood[151], Lynn M. Yee[152], Natalie M. Young[19], Sokratis N. Zisis[97], Andrea S. Foulkes[2]**

**1** Department of Population Health, NYU Grossman School of Medicine, New York, New York, United States of America, **2** Department of Biostatistics, Massachusetts General Hospital, Boston, Massachusetts, United States of America, **3** Division of Pulmonary Critical Care and Sleep Medicine, NYU Langone Health, New York, New York, United States of America, **4** Department of Laboratory Medicine and Pathology, Mayo Clinic, Rochester, Minnesota, United States of America, **5** Patient Led Research Collaboration on COVID-19, Washington, DC, United States of America, **6** Division of Infectious Diseases, Providence Swedish Medical Center, Seattle, Washington, United States of America, **7** Department of Population Health Sciences and Internal Medicine, University of Utah School of Medicine, Salt Lake City, Utah, United States of America, **8** Department of Medicine, West Virginia University, Morgantown, West Virginia, United States of America, **9** Department of Obstetrics, Gynecology, and Reproductive Sciences, University of California, San Francisco, San Francisco, California, United States of America, **10** Division of Geographic Medicine and Infectious Diseases, Tufts Medical Center, Medford, Massachusetts, United States of America, **11** Department of Medicine, University of Illinois Chicago, Chicago, Illinois, United States of America, **12** Department of Medicine, Howard University, Washington, DC, United States of America, **13** Department of Obstetrics and Gynecology, University of Utah Health, Salt Lake City, Utah, United States of America, **14** Body Politic COVID-19 Support Group, Boston, Massachusetts, United States of America, **15** Department of Medicine and Surgery, Health Services Research Center, Emory University School of Medicine, Atlanta, Georgia, United States of America, **16** Department of Medicine and Rheumatology, The MetroHealth Medical Center, Cleveland, Ohio, United States of America, **17** Department of Neurology, Stony Brook University Renaissance School of Medicine, Stony Brook, New York, United States of America, **18** Department of Medicine, Division of Infectious Diseases and Infectious Diseases, Long School of Medicine, University of Texas Health Science Center San Antonio, San Antonio, Texas, United States of America, **19** Swedish Center for Research and Innovation, Providence Swedish Medical Center, Seattle, Washington, United States of America, **20** Department of Epidemiology and Biostatistics, University of California at San Francisco Institute of Global Health Sciences, San Francisco, San Francisco, California, United States of America, **21** College of Science and Health, Department of Health Sciences, DePaul University, Chicago, Illinois, United States of America, **22** Medical Practice Evaluation Center, Department of Medicine, Massachusetts General Hospital, Harvard Medical School, Boston, Massachusetts, United States of America, **23** Department of Medicine, Emory University, Atlanta, Georgia, United States of America, **24** Center for Emerging Infectious Diseases Policy and Research, Boston University School of Medicine, Boston, Massachusetts, United States of America, **25** Department of Medicine, University of Arizona, Tucson, Arizona, United States of America, **26** Department of Clinical Research, Sanford Research, Sioux Falls, South Dakota, United States of America, **27** Department of Emergency Medicine, Stanford University, Stanford, California, United States of America, **28** Department of Pathology, Howard University, Washington, DC, United States of America, **29** Division of Pulmonary and Critical Care, University of Kansas Medical Center, Kansas City, Kansas City, United States of America, **30** Charles Bronfman Institute for Personalized Medicine, Icahn School of Medicine at Mount Sinai, New York, New York, United States of America, **31** Division of Infectious Disease, Department of Medicine, Icahn School of Medicine at Mount Sinai, New York, New York, United States of America, **32** Department of Medicine, Brigham and Women's Hospital, Boston, Massachusetts, United States of America, **33** Department of Medicine, Cedars-Sinai Medical Center, Los Angeles, California, United States of America, **34** Department of Emergency Medicine, University of Illinois Chicago, Chicago, Illinois, United States of America, **35** Department of Neurology, Massachusetts General Hospital, Boston, Massachusetts, United States of America, **36** Department of Medicine, University of Hawaii at Manoa John A. Burns School of Medicine, Honolulu, Hawaii, United States of America, **37** Department of Allergy & Infectious Diseases, University of Washington, Seattle, Washington, United States of America, **38** Department of Biostatistics, George Washington University, Washington, DC, United States of America, **39** Department of Obstetrics and Gynecology, The Ohio State University Hospital, Columbus, Ohio, United States of America, **40** Department of Medicine, University of California, San Francisco, San Francisco, California, United States of America, **41** Department of Internal Medicine, University of Nebraska Medical Center, Omaha, Nebraska, United States of America, **42** Department of Research Services, University of Illinois College of Medicine, Peoria, Illinois, United States of America, **43** Department of Medicine, Division of Cardiology at Zuckerberg San Francisco General, University of California San Francisco, San Francisco, California, United States of America, **44** MaineHealth Institute for Research, MaineHealth, Scarborough, Maine, United States of America, **45** Department of Medicine, University of Colorado Anschutz Medical Campus, Aurora, Colorado, United States of America, **46** Department of Biomedical Informatics and Clinical and Translational Science Institute, University of Utah, Salt Lake City, Utah, United States of America, **47** Department of Medicine, Division of Infectious Diseases, Emory University School of Medicine, Atlanta, Georgia, United States of America, **48** Department of Pathology, CVPath Institute, Gaithersburg, Maryland, United States of America, **49** Department of Pediatrics,

UpToDate and serving on a Data Safety Monitoring board for Kintor Pharmaceuticals, Ltd. Bruce Levy reported serving as a consultant for AstraZeneca, Entrinsic Biosciences, Gossamer Bio and Nocion Therapeutics and receiving research support from Amgen, Genentech, GlaxoSmithKline, Pieris Pharmaceuticals, SRA and Sanofi unrelated to the current work. Vincent Marconi reported receiving grants from NIH during the conduct of the study and grants from NIH, Veteran Affairs, and Centers for Disease Control and Prevention; grants, personal fees, nonfinancial support, and other from Lilly and Gilead; grants and personal fees from ViiV; and nonfinancial support from Bayer outside the submitted work. Grace McComsey reported serving as consultant for Merck, Gilead, ViiV, Janssen and have received research support from Pfizer, Vanda, Genentech, Roche, Redhill and Cognivue. Torri Metz reported being a site PI and a participant in the medical advisory board for the planning of a Pfizer clinical trial of SARS-CoV-2 vaccination in pregnancy. She also reported being a site PI for a Pfizer study evaluating the pharmacokinetics of Paxlovid in pregnant people with COVID-19. Janet Mullington reported support for investigator-initiated research by "Open Medicine Foundation and the Patient-Led Research Collaborative" Princess Ogbogu reported research support from Astrazeneca, GSK, Blueprint medical; advisory board for Astrazeneca, GSK, Sanofi, Kalvista; and consulting for Astrazeneca, GSK Sairam Parthasarathy reported research funding to Institution from Sergey Brin foundation of COVID and Long-COVID research. Michael Peluso reported consulting fees from Gilead Sciences and AstraZeneca, and service on data safety monitoring board for American Gene Technologies. Sean Quigley reported service on speaker Board for Servier, Alnylam, Agios; service on advisory board for Recordati, Alexion. Franz Rischard reported research support from NIH/NHLBI, United Therapeutics, Acceleron/Merck, Janssen, Insmed, Aerovate, and Bayer; and consulting/advisory compensation from Acceleron and Bayer. Nadine Rouphael reported being a consultant for ICON and EMMES as a safety consultant for clinical trials; service on the advisory boards for Moderna; funds to institution from Sanofi, Lilly, Merck, Quidel and Pfizer. PJ Utz reported stock ownership of Gilead and PI of biomarker studies for Pfizer STOP-PASC paxlovid trial Juan Wisnivesky received receiving consulting honorarium from Sanofi, Banook, Prospero, PPD and Atea and research grants from Sanofi, Regeneron, Axella, and Arnold Consultants.

University of California, San Francisco, San Francisco, California, United States of America, 50 Mile Square Health Center, University of Illinois Chicago, University of Illinois Chicago, Chicago, Illinois, United States of America, 51 Department of Medicine, Tulane University School of Medicine, New Orleans, Louisiana, United States of America, 52 Department of Endocrinology, Diabetes and Bone Disease, Icahn School of Medicine at Mount Sinai, New York, New York, United States of America, 53 Center for Research and Evaluation, Kaiser Permanente of Georgia, Atlanta, Georgia, United States of America, 54 Public Health Research Institute and Department of Medicine, Rutgers New Jersey Medical School, Newark, New Jersey, United States of America, 55 Department of Obstetrics and Gynecology, MetroHealth System, Cleveland, Ohio, United States of America, 56 Department of Medicine, Stanford University, Stanford, California, United States of America, 57 Department of Epidemiology and Population Health, Stanford University School of Medicine, Stanford, California, United States of America, 58 Department of Physiology & Biophysics, University of Mississippi Medical Center, Jackson, Mississippi, United States of America, 59 Clinical Trials, Pennington Biomedical Research Center, Baton Rouge, Louisiana, United States of America, 60 Department of Emergency Medicine, OSF Saint Francis Medical Center, Peoria, Illinois, United States of America, 61 Department of Pulmonary and Critical Care, Emory University School of Medicine, Atlanta, Georgia, United States of America, 62 Department of Internal Medicine University of New Mexico, Health Science Center, Albuquerque, New Mexico, United States of America, 63 Clinical Research Center, Beth Israel Deaconess Medical Center, Boston, Massachusetts, United States of America, 64 Department of Bioengineering, Institute for Systems Biology, Seattle, Washington, United States of America, 65 Clinical Research Center, University Hospitals Cleveland Medical Center, Cleveland, Ohio, United States of America, 66 Seattle Children's Therapeutics, Seattle, Washington, United States of America, 67 Department of Obstetrics and Gynecology, Christiana Care Health Services, Newark, Delaware, United States of America, 68 Institute for Health Equity Research, Icahn School of Medicine at Mount Sinai, New York, New York, United States of America, 69 Department of Internal Medicine, University of Arizona, Phoenix, Arizona, United States of America, 70 Department of Obstetrics and Gynecology, Duke University, Durham, North Carolina, United States of America, 71 Department of Arthritis & Clinical Immunology, Oklahoma Medical Research Foundation, Oklahoma City, Oklahoma, United States of America, 72 Department of Family Medicine, Cambridge Health Alliance, Cambridge, Massachusetts, United States of America, 73 Department of Pulmonary and Critical Care Medicine, University of Colorado, Aurora, Colorado, United States of America, 74 Department of Biostatistics, University of Alabama at Birmingham, Birmingham, Alabama, United States of America, 75 Department of Medicine, Division of Infectious Diseases, West Virginia School of Medicine, Morgantown, West Virginia, United States of America, 76 Department of Infectious Diseases, Massachusetts General Hospital, Boston, Massachusetts, United States of America, 77 Department of Medicine, Harvard Medical School, Boston, Massachusetts, United States of America, 78 Department of Medicine, NYU Langone Health, New York, New York, United States of America, 79 Department of Pediatrics & Department of Research Services, University of Illinois College of Medicine, Peoria, Illinois, United States of America, 80 Department Integrated Physiology and Molecular Medicine, Pennington Biomedical Research Center, Baton Rouge, Louisiana, United States of America, 81 Department of Psychiatry, Harvard Medical School, Boston, Massachusetts, United States of America, 82 Department of Internal Medicine and Neurology, University of Arizona College of Medicine Phoenix, Phoenix, Arizona, United States of America, 83 Communications Practice Area, RTI International, Research Triangle Park, North Carolina, United States of America, 84 Department of Internal Medicine, Brigham and Women's Hospital, Boston, Massachusetts, United States of America, 85 Tulane Center for Clinical Research, Tulane University School of Medicine, New Orleans, Louisiana, United States of America, 86 Department of Medicine, University of Washington, Seattle, Washington, United States of America, 87 Tulane Center for Clinical Neurosciences, Tulane School of Medicine, New Orleans, Louisiana, United States of America, 88 Department of Neurology, University of Puerto Rico School of Medicine, San Juan, Puerto Rico, United States of America, 89 Department of Family & Community Medicine, University of Arizona, College of Medicine – Tucson, Tucson, Arizona, United States of America, 90 Department of Medicine, Case Western Reserve University School of Medicine, Cleveland, Ohio, United States of America, 91 Department of Obstetrics and Gynecology, Northwestern University, Chicago, Illinois, United States of America, 92 Department of Medicine, Section of Infectious Diseases, Boston University Medical Center, Boston, Massachusetts, United States of America, 93 Department of Medicine, Infectious Diseases and Department of Global Health, Emory University School of Medicine, Atlanta, Georgia, United States of America, 94 Department of Medicine, University of Mississippi Medical Center, Jackson, Mississippi, United States of America, 95 Department of Epidemiology and Biostatistics, University of California, San Francisco, San Francisco, California, United States of America, 96 Department of Cardiology, Intermountain Medical Center, Salt Lake City, Utah, United States of America, 97 Department of Medicine, Department of Pediatrics, Case Western Reserve University School of Medicine, Cleveland, Ohio, United States of America, 98 Department of Obstetrics, Gynecology and Reproductive Sciences, University of Texas Health Science Center at Houston, Houston, Texas, United States of America, 99 Department of Genetics, Louisiana State University Health Sciences Center, New Orleans, Louisiana, United States of America, 100 Department of Epidemiology, Harvard T.H. Chan School of Public Health, Boston, Massachusetts, United States of America, 101 Department of Infectious Diseases, Children's

Hospital Los Angeles, University of Southern California, Los Angeles, California, United States of America, **102** Department of Neurology and Sleep Medicine, Beth Israel Deaconess Medical Center, Boston, Massachusetts, United States of America, **103** Division of General Internal Medicine, Icahn School of Medicine at Mount Sinai, New York, New York, United States of America, **104** Division of Cardiology, Kaiser Permanente of Georgia, Atlanta, Georgia, United States of America, **105** Department of Social, Statistical and Environmental Sciences, RTI International, Research Triangle Park, North Carolina, United States of America, **106** Department of Immunobiology, University of Arizona College of Medicine, Tucson, Arizona, United States of America, **107** Division of Pediatric Allergy, Immunology, and Rheumatology, University Hospitals Rainbow Babies and Children's Hospital, Cleveland, Ohio, United States of America, **108** Department of Obstetrics and Gynecology, Medical College of Wisconsin, Milwaukee, Wisconsin, United States of America, **109** Department of Obstetrics and Gynecology, Division of Maternal Fetal Medicine, Saint Peter's University Hospital, New Brunswick, New Jersey, United States of America, **110** Department of Obstetrics and Gynecology, University of Pennsylvania, Philadelphia, Pennsylvania, United States of America, **111** Department of Medicine, Department of Infectious Disease, University of Texas Health, San Antonio, Texas, United States of America, **112** Division of HIV, Infectious Disease, and Global Medicine, University of California, San Francisco, California, United States of America, **113** Department of Medicine, Morehouse School of Medicine, Atlanta, Georgia, United States of America, **114** Department of Obstetrics, Gynecology & Reproductive Sciences, Yale School of Medicine, New Haven, Connecticut, United States of America, **115** Department of Obstetrics and Gynecology, NorthShore University Health System, Evanston, Illinois, United States of America, **116** Department of Epidemiology and Biostatistics, University of Arizona, Tucson, Arizona, United States of America, **117** Division of Cardiology, Warren Alpert Medical School of Brown University, Providence, Rhode Island, United States of America, **118** Department of Internal Medicine, Division of Infectious Diseases, University of Kentucky, Lexington, Kentucky, United States of America, **119** Department of Medicine, Division of Hematology/Oncology, University of Illinois Chicago, Chicago, Illinois, United States of America, **120** Department of Psychiatry and Behavioral Sciences, University of New Mexico School of Medicine, Albuquerque, New Mexico, United States of America, **121** Department of Pediatrics, University of New Mexico, Health Sciences Center, Albuquerque, New Mexico, United States of America, **122** Department of Nutrition and Chronic Disease, Pennington Biomedical Research Center, Baton Rouge, Louisiana, United States of America, **123** Department of Obstetrics and Gynecology, Columbia University, New York, New York, United States of America, **124** Department of Pulmonary and Critical Care, University of Arizona, Tucson, Arizona, United States of America, **125** Department of Obstetrics and Gynecology, Brown University, Providence, Rhode Island, United States of America, **126** Department of Medicine, University of Puerto Rico, San Juan, Puerto Rico, United States of America, **127** Department of Internal Medicine, University of Utah, Salt Lake City, Utah, United States of America, **128** Department of Medicine, Division of Pulmonary Allergy and Critical Care Medicine, University of Pittsburgh, Pittsburgh, Pennsylvania, United States of America, **129** Glenn Biggs Institute for Alzheimer's and Neurodegenerative Diseases, University of Texas Health Sciences Center San Antonio, San Antonio, Texas, United States of America, **130** Division of Preventive Medicine, Brigham and Women's Hospital, Boston, Massachusetts, United States of America, **131** Department of Population Health Sciences, Mays Cancer Center, University of Texas Health, San Antonio, Texas, United States of America, **132** Department of Pediatrics, Icahn School of Medicine at Mount Sinai, New York, New York, United States of America, **133** Department of Biochemistry & Molecular Biology, Howard University College of Medicine, Washington, DC, United States of America, **134** Department of Obstetrics, Gynecology, and Reproductive Sciences, University of Pittsburgh School of Medicine, Pittsburgh, Pennsylvania, United States of America, **135** Department of Internal Medicine, Stanford University, Stanford, California, United States of America, **136** Department of Biomedical Engineering, Department of Systems and Industrial Engineering, University of Arizona College of Engineering, Tucson, Arizona, United States of America, **137** Department of Pediatrics, Emory Vaccine Center, Emory University, Atlanta, Georgia, United States of America, **138** Health Services Research Center, Emory University, Atlanta, Georgia, United States of America, **139** Department of Obstetrics and Gynecology, University of North Carolina Chapel Hill, Chapel Hill, North Carolina, United States of America, **140** Department of Obstetrics and Gynecology and Center for Women's Reproductive Health, University of Alabama at Birmingham, Birmingham, Alabama, United States of America, **141** Department of Medicine, University of Texas Health Science Center, San Antonio, Texas, United States of America, **142** Department of Medicine, Division of Nephrology, University of Washington School of Medicine, Spokane, Washington, United States of America, **143** Department of Pediatrics, University of Oklahoma Health Sciences Center, Oklahoma City, Oklahoma, United States of America, **144** Department of Research, Bateman Horne Center, Salt Lake City, Utah, United States of America, **145** Department of Medicine, Tufts Medical Center, Boston, Massachusetts, United States of America, **146** Department of Neurological Sciences, University of Nebraska Medical Center, Omaha, Nebraska, United States of America, **147** Department of Microbiology, Immunology, and Molecular Genetics, University of Kentucky, Lexington, Kentucky, United States of America, **148** Center for Individualized Medicine, Mayo Clinic, Rochester, Minnesota, United States of America, **149** Department of Molecular and Cellular Biochemistry, University of Kentucky, Lexington, Kentucky, United States of America, **150** Institute for Excellence in Health Equity, NYU Grossman School of Medicine, New York, New York,

United States of America, **151** Department of Pediatrics and Radiology, Children's Hospital of Los Angeles, Los Angeles, California, United States of America, **152** Department of Obstetrics and Gynecology, Northwestern University Feinberg School of Medicine, Chicago, Illinois, United States of America

\* Leora.Horwitz@nyulangone.org

# Abstract

### Importance

SARS-CoV-2 infection can result in ongoing, relapsing, or new symptoms or other health effects after the acute phase of infection; termed post-acute sequelae of SARS-CoV-2 infection (PASC), or long COVID. The characteristics, prevalence, trajectory and mechanisms of PASC are ill-defined. The objectives of the Researching COVID to Enhance Recovery (RECOVER) Multi-site Observational Study of PASC in Adults (RECOVER-Adult) are to: (1) characterize PASC prevalence; (2) characterize the symptoms, organ dysfunction, natural history, and distinct phenotypes of PASC; (3) identify demographic, social and clinical risk factors for PASC onset and recovery; and (4) define the biological mechanisms underlying PASC pathogenesis.

### Methods

RECOVER-Adult is a combined prospective/retrospective cohort currently planned to enroll 14,880 adults aged ≥18 years. Eligible participants either must meet WHO criteria for suspected, probable, or confirmed infection; or must have evidence of no prior infection. Recruitment occurs at 86 sites in 33 U.S. states, Washington, DC and Puerto Rico, via facility- and community-based outreach. Participants complete quarterly questionnaires about symptoms, social determinants, vaccination status, and interim SARS-CoV-2 infections. In addition, participants contribute biospecimens and undergo physical and laboratory examinations at approximately 0, 90 and 180 days from infection or negative test date, and yearly thereafter. Some participants undergo additional testing based on specific criteria or random sampling. Patient representatives provide input on all study processes. The primary study outcome is onset of PASC, measured by signs and symptoms. A paradigm for identifying PASC cases will be defined and updated using supervised and unsupervised learning approaches with cross-validation. Logistic regression and proportional hazards regression will be conducted to investigate associations between risk factors, onset, and resolution of PASC symptoms.

### Discussion

RECOVER-Adult is the first national, prospective, longitudinal cohort of PASC among US adults. Results of this study are intended to inform public health, spur clinical trials, and expand treatment options.

### Registration

NCT05172024.

## Introduction

Hundreds of millions of people worldwide have been infected with the Severe Acute Respiratory Syndrome Coronavirus-2 (SARS-CoV-2) [1]. Many have experienced ongoing, relapsing, or new symptoms or other health effects occurring after the acute phase of infection, termed post-acute sequelae of SARS-CoV-2 infection (PASC), or long COVID. While more than 200 symptoms have been associated with PASC [2], there are no agreed upon criteria for the diagnosis of PASC, and estimates of PASC incidence and prevalence vary widely [3–9].

The pathophysiology underlying PASC remains incompletely understood [10, 11]. Various mechanisms have been proposed, including viral persistence [12–17], microvascular clotting and platelet dysregulation [18–21], tissue damage from initial infection [22, 23], inflammation and immune dysregulation [17, 24–31], reactivation of other latent viral infections (e.g., Epstein-Barr virus) [17, 32, 33], microbial translocation and dysbiosis [34, 35], and/or impacts of pandemic-related disruptions on health [36–38]. Further characterization of PASC clinical manifestations and underlying pathophysiologic mechanisms could facilitate identification and investigation of preventive and therapeutic interventions.

## Materials and methods

### Objectives

The National Institutes of Health (NIH) initiative "Researching COVID to Enhance Recovery (RECOVER) Multi-site Observational Study of Post-Acute Sequelae of SARS-CoV-2 Infection in Adults" (RECOVER-Adult) is intended to: (1) characterize the incidence and prevalence of PASC; (2) characterize the spectrum of clinical symptoms, subclinical organ dysfunction, natural history, and distinct phenotypes identified as PASC; (3) identify demographic, social determinants of health (SDoH), and clinical risk factors for PASC and PASC recovery, and (4) define the biological mechanisms underlying pathogenesis of PASC. This report describes the study design of the RECOVER-Adult study.

### Study design

RECOVER is an ambidirectional (combined retrospective and prospective) longitudinal cohort study that includes people infected or uninfected with SARS-CoV-2. Participants may be enrolled at the time of SARS-CoV-2 infection or a negative test (for uninfected group) and followed prospectively; or, may be enrolled after SARS-CoV-2 infection or a negative test, asked retrospectively about symptoms since infection or a negative test, and then followed prospectively. Participants may be followed until October 2025. An embedded cohort study of pregnant people will provide longitudinal follow-up of the birthing parent and offspring. The protocol for the pregnancy cohort is reported separately [39]. In addition, a series of nested case-control studies will be performed among participants with and without select symptoms or findings, who will undergo more intensive radiographic imaging, physiologic assessment, and tissue collection.

### Protocol development

The protocol was developed and refined in a collaborative process involving site and core investigators, patient representatives and caregivers, and NIH staff (see S1 Fig for timeline and details). Patient representatives were recruited from COVID advocacy organizations such as the Patient-Led Research Collaborative, Survivor Corps, and Long COVID Families; patient organizations with expertise in post-viral syndromes; grass-roots activist organizations; and through nominations by enrolling sites. All patient representatives were compensated for their

time. Refinements to the protocol continue to be made in response to participant and site feedback, new scientific evidence, and interim results.

## Study organizational structure and study management

The study infrastructure includes four cores: (1) the Clinical Science Core (CSC) at New York University (NYU) Grossman School of Medicine oversees study sites and provides scientific leadership in collaboration with the site Principal Investigators, (2) the Data Resource Core (DRC) at Massachusetts General Hospital and Brigham and Women's Hospital provides scientific and statistical leadership, and handles data management and storage, (3) the PASC Biorepository Core (PBC) at Mayo Clinic manages biospecimens obtained from study sites, and (4) the Administrative Coordinating Center (ACC) at RTI International (RTI) provides operational and administrative support; collectively these form the Core Operations Group. The four cores are supported by six Oversight Committees that oversee RECOVER-wide activities including publications, ancillary studies, clinical trial interventions selection, quality assurance, and study design. Twelve pathobiology task forces provide content-specific input. All RECOVER cohort studies receive inputs from the National Community Engagement Group composed of patient and community representatives; and are overseen by a Steering Committee composed of core and hub principal investigators, patient representatives and NIH program leadership; an Executive Committee composed of NIH Institute leaders, patient representatives and other federal leadership; and an Observational Study Monitoring Board (OSMB) (S2 Fig) [40].

## Study setting and participating sites

RECOVER-Adult is designed as a hub and spoke model, with 16 hubs collectively overseeing 86 enrolling sites in the United States (U.S.) located in 33 states plus Washington, DC and Puerto Rico (S1 Table). Enrolling sites include hospitals, health centers, and community organizations drawing participants primarily from their surrounding communities. Two sites are mobile health vans enrolling in rural communities far from health centers. One hub is enrolling participants remotely across the country, with study procedures conducted through home visits and biospecimen collection at local laboratories.

## Eligibility criteria

Participants are eligible for RECOVER if they are at least 18 years old, have reached the age of majority in their state of residence, are not incarcerated, and are not terminally ill. Individuals with or without history of SARS-CoV-2 infection are eligible. Infected individuals must have suspected, probable, or confirmed SARS-CoV-2 infection as defined by World Health Organization (WHO) criteria [41] (see Table 1), or positive SARS-CoV-2 infection-specific antibody testing. Uninfected individuals must not meet any WHO criteria for infection and must have a documented negative SARS-CoV-2 nucleic acid and antibody test result (Table 1).

Individuals who were pregnant at the time of a SARS-CoV-2 infection and had a live birth, or who are pregnant at the time of enrollment in RECOVER, are only eligible to enroll in the pregnancy cohort of the adult study. Their offspring are eligible for enrollment into the congenital exposure cohort of the RECOVER pediatric study. Individuals who were pregnant at the time of a SARS-CoV-2 infection and had a pregnancy loss or termination prior to 20 weeks' gestation, are eligible to enroll in either the pregnancy or the adult main cohort.

**Table 1. Definition of infected and uninfected categories used in RECOVER-Adult.** Changes from WHO definition are indicated in italics.

| WHO category | Criteria |
| --- | --- |
| Suspected | Acute onset of fever and cough OR acute onset of any three of more of the following signs or symptoms: fever, cough, general weakness/fatigue, headache, myalgia, sore throat, coryza, dyspnea, anorexia/nausea/vomiting, diarrhea, altered mental status. |
| | AND at least one of: 1. Residing or working in an area with a high risk of transmission of virus: closed residential settings, humanitarian settings such as camp and camp-like settings for displaced persons; anytime within the 14 days before symptom onset; OR 2. Residing or travel to an area with community transmission* anytime within the 14 days before symptom onset; OR 3. Working in any health care setting, including within health facilities and within households or within the community; anytime within the 14 days before symptom onset. |
| | *AND* |
| | *Did not have a negative test for SARS-CoV-2 at the time of suspected infection.* |
| | Severe acute respiratory illness: acute respiratory infection with history of fever or measured fever of ≥38C˚; and cough; with onset within the last 10 days; and requires hospitalization |
| | *AND* |
| | *Did not have a negative test for SARS-CoV-2 at the time of suspected infection.* |
| | A positive SARS-CoV-2 Antigen-RDT who is asymptomatic *or meets some but not all clinical or epidemiologic criteria* |
| | *AND* |
| | *Did not have a negative test for SARS-CoV-2 at the time of suspected infection.* |
| Probable | A patient who meets clinical criteria for suspected SARS-CoV-2 AND is a contact of a probable or confirmed case or linked to a COVID-19 cluster |
| | *AND* |
| | *Did not have a negative test for SARS-CoV-2 at the time of suspected infection.* |
| | A patient who meets clinical criteria for suspected SARS-CoV-2 AND has chest imaging showing findings suggestive of COVID-19 disease |
| | *AND* |
| | *Did not have a negative test for SARS-CoV-2 at the time of suspected infection.* |
| | A person with recent onset of anosmia (loss of smell) or ageusia (loss of taste) in the absence of any other identified cause |
| | *AND* |
| | *Did not have a negative test for SARS-CoV-2 at the time of suspected infection.* |
| Confirmed | Any person with a positive Nucleic Acid Amplification Test (NAAT) |
| | Any person with a positive SARS-CoV-2 Antigen-RDT *(including home-administered rapid test)* AND meeting either the probable case definition or one of the first two suspected criteria |
| | An asymptomatic person with a positive SARS-CoV-2 Antigen-RDT (including home-administered rapid test) who is a contact of a probable or confirmed case |
| | *Any person with a positive SARS-CoV-2 nucleocapsid protein antibody test OR a positive SARS-CoV-2 spike protein antibody test IF not vaccinated* |
| Uninfected | Does not meet WHO criteria for a suspected, probable, or confirmed case of SARS-CoV-2 infection |
| | AND |
| | Has negative NAAT SARS-CoV-2 testing from a respiratory specimen performed at the time of enrollment/screening |
| | AND |
| | Has a negative SARS-CoV-2 nucleocapsid protein antibody and spike protein antibody test (if not vaccinated) performed at the time of enrollment |
| | AND |
| | Lives in the same communities or recruited from the same sources as those in the SARS-CoV-2 infected cohort |

## Sample size

Sample size determinations were performed for both aggregate and subgroup analyses based on characteristics such as age, sex, race/ethnicity, pregnancy, and vaccination status. The current version of the protocol targets enrollment of 12,200 participants with history of SARS-CoV-2 infection and 2,680 participants without history of SARS-CoV-2 infection (total of 14,880). Sample size targets are further specified by duration of time between infection (or negative test) and enrollment, and by pregnancy status (Table 2). Based on 90% power and a type-1 error rate of 0.01, the minimum detectable effect size for the difference in risk of PASC or a PASC symptom between participants with and without infection is 3.1% (6.4% in a 25% subgroup), assuming the risk among participants without infection is 15%. When restricting to acute infected and uninfected participants only, the minimum detectable risk difference is 4.7% (10.0% in a 25% subgroup). In logistic regression analyses investigating whether an infected participant develops PASC, the minimum detectable odds ratio for a risk factor is 1.22 (1.46 in a 25% subgroup), assuming 25% of all infected participants develop PASC and that the risk factor prevalence among participants who do not develop PASC is 20%. Finally, assuming that 50% of infected participants who develop PASC recover from it during follow-up, the minimum detectable odds ratio for the association between a risk factor and recovering from PASC is 1.40 (1.90 in a 25% subgroup), assuming the risk factor has 20% prevalence among participants who do not recover from PASC.

Sample size targets for race/ethnicity are intended to match the distribution of SARS-CoV-2 infection in the U.S. as of June 2021: 16% non-Hispanic Black; 27% Hispanic, 4% non-Hispanic Asian, American Indian/Alaska Native or Native Hawaiian/Other Pacific Islander; and 53% non-Hispanic White [42].

## Recruitment

Participants are recruited through outreach to patients cared for at the enrolling site, community outreach, use of public health test lists, and self-referrals from the RECOVER website (https://recovercovid.org). For participants without SARS-CoV-2 infection, sites are asked to draw from similar communities, demographics, and sites of care as those recruiting infected

**Table 2. Sample size targets, by enrollment category.**

| Enrollment category | Target sample size, by subgroup | Target sample size, by time since infection |
|---|---|---|
| Infected, enrolled within 30 days of infection, not pregnant ("acute" infected) | 4,714 | 5,000 |
| Infected, enrolled within 30 days of infection, pregnant at time of infection ("acute" infected) | 286 | |
| Infected, enrolled >30 days after infection, not pregnant ("post-acute" infected) | 5,619 | 7,200 |
| Infected, enrolled >30 days after infection, pregnant at time of infection ("post-acute" infected | 1,581 | |
| Uninfected, enrolled within 30 days of negative test, not pregnant ("acute" uninfected) | 1,141 | 1,200 |
| Uninfected, enrolled within 30 days of negative test, pregnant ("acute" uninfected) | 59 | |
| Uninfected, enrolled >30 days from negative test, not pregnant ("post-acute" uninfected) | 1,106 | 1,480 |
| Uninfected, enrolled >30 days from negative test, pregnant ("post-acute" uninfected) | 374 | |

participants. Enrollment is tracked by enrollment category, race/ethnicity, sex, residence in rural or medically underserved areas, hospitalization status at time of initial infection, and referral source to allow for real time adjustments in enrollment to match protocol targets.

## Assessments

The RECOVER-Adult schedule of assessments includes: surveys, collection of biologic specimens, physical examinations, laboratory tests, radiologic studies, and invasive procedures to measure study outcomes. The schedule starts at the time of first infection or at the negative test date ("index date"); follow-up visits are conducted at 90-day intervals for a maximum of 4 years. All participants undergo the same assessments at baseline enrollment. Thereafter, participants follow the assessment schedule corresponding to the appropriate time point relative to the index date. For example, participants who are enrolled 90 days after infection follow the 180-day assessment schedule at their first follow-up visit (Fig 1). Participants may remain in the study if they have missed a visit; after three missed visits they may be considered lost to follow-up, in which case no further information is obtained.

**Surveys.** Participants complete surveys at 90-day intervals throughout the study. On enrollment, data are collected on demographics, SDoH, disability, characteristics of the initial SARS-CoV-2 infection (if applicable), pregnancy (if applicable), vaccination status, comorbidities, medications, and PASC symptoms. Subsequently, at 90-day intervals, data are collected on interim infections, time-varying social determinants, vaccinations, comorbidities, medications and symptoms. The PASC symptom survey was developed for RECOVER and includes an overall quality of life instrument (PROMIS-10) and screening for core symptoms (43 for biological males and 46 for biological females) drawn from existing literature plus input from patient representatives and investigators. Questions about depression, anxiety, post-traumatic stress disorder (PTSD), and grief are also included. Report of a symptom may trigger additional questions about that symptom. Wherever possible, pre-existing validated survey instruments are used. Details of survey instruments can be found at https://recovercovid.org/protocols and in S2 Table.

| eCRF | Baseline | 3m | 6m | 9m | 12m | 15m | 18m | 21m | 24m | 27m | 30m | 33m | 36m | 39m | 42m | 45m | 48m |
|---|---|---|---|---|---|---|---|---|---|---|---|---|---|---|---|---|---|
| | | | | | | | | | Time Point after index date | | | | | | | | |
| Enrollment | ● | | | | | | | | | | | | | | | | |
| Tier 1-2 Consent | ● | | | | | | | | | | | | | | | | |
| Identity | ● | | | | | | | | | | | | | | | | |
| Visit | ● | ● | ● | ● | ● | ● | ● | ● | ● | ● | ● | ● | ● | ● | ● | ● | ● |
| Comorbidities | ● | ● | ● | ● | ● | ● | ● | ● | ● | ● | ● | ● | ● | ● | ● | ● | ● |
| COVID Treatment* | ● | | | | | | | | | | | | | | | | |
| Medications | ● | | | | | | | | | | | | | | | | |
| Change in Medications | | ● | ● | ● | ● | ● | ● | ● | ● | ● | ● | ● | ● | ● | ● | ● | ● |
| Demographics | ● | | | | | | | | | | | | | | | | |
| PASC Symptoms | ● | ● | ● | ● | ● | ● | ● | ● | ● | ● | ● | ● | ● | ● | ● | ● | ● |
| Vaccination Status | ● | ● | ● | ● | ● | ● | ● | ● | ● | ● | ● | ● | ● | ● | ● | ● | ● |
| Social Determinants of Health | ● | | | | | | | | | | | | | | | | |
| Social Determinants Follow-up | | ● | ● | ● | ● | ● | ● | ● | ● | ● | ● | ● | ● | ● | ● | ● | ● |
| Alcohol/Tobacco | ● | | | | | | | | | | | | | | | | |
| Alcohol/Tobacco Follow-up | | ● | ● | ● | ● | ● | ● | ● | ● | ● | ● | ● | ● | ● | ● | ● | ● |
| Disability | ● | | | | | | | | | | | | | | | | |
| Pregnancy | ● | | | | | | | | | | | | | | | | |
| Pregnancy Follow-up | | ● | ● | ● | ● | ● | ● | ● | ● | ● | ● | ● | ● | ● | ● | ● | ● |
| Tier 1 office visit | ● | | ● | | ● | | | | ● | | | | ● | | | | ● |
| Biospecimens | ● | ● | ● | | ● | | | | ● | | | | ● | | | | ● |
| Lab Results | ● | ● | ● | | ● | | | | ● | | | | ● | | | | ● |
| Tier 2/Tier 3 Tests | | | | | | | | | | | | | | | | | |

\* COVID Treatment not collected on people without infection

**Legend**
● Completed by research coordinator
● Completed by participant
● Completed by research coordinator with review/validation by participant

**Fig 1. Schedule of assessments.**

**In-person assessments.** Office-based assessments are performed on all participants at enrollment, at 180 days after index date, and then at yearly increments thereafter. These include: height, weight, waist circumference, seated vitals (heart rate, blood pressure, oxygen saturation), 30 second sit-to-stand, and a 10 minute active stand test during which heart rate and blood pressure are measured at 1, 3, 5 and 10 minute intervals (S3 Table).

**Laboratory assessments.** A core set of laboratory studies are obtained on all participants at enrollment, at 90 and 180 days after the index date (S3 Table). After 180 days, abnormal laboratory tests from the most recent prior visit are repeated annually. At enrollment only, participants enrolled as uninfected undergo SARS-CoV-2 PCR testing and SARS-CoV-2 antibody testing (nucleocapsid for all, and spike only for unvaccinated). These core studies are performed at each site in Clinical Laboratory Improvement Amendments of 1988 (CLIA)-certified laboratories.

**Biospecimens.** At enrollment, at 90 and 180 days after the index date, and then annually, participants are asked to provide blood and nasopharyngeal/nasal swab biospecimens for storage (Table 3). Saliva is collected once upon enrollment for genetic analysis. Urine and stool are collected biannually. Biospecimens are not collected from participants who decline use of samples for future research.

**Triggered testing.** Participants with infection experiencing specific symptoms or having abnormal study assessments may be eligible for additional assessments, each of which is triggered by qualifying criteria. In addition, participants with and without infection are randomly selected to complete additional assessments for comparison. These assessments are divided into Tier 2 and Tier 3 assessments. Tier 2 assessments are anticipated to be completed by approximately 30% of participants per assessment, and may be repeated yearly if abnormal (S4 Table). Tier 3 assessments are more invasive and/or burdensome, and are anticipated to be completed by not more than 20% of participants per assessment. Tier 3 assessments with more than minimal risk can only be performed once (S5 Table). Participants are eligible to begin Tier 2 and Tier 3 testing 90 to 180 days after index date, depending on the assessment.

**Table 3. Tier 1 biospecimen collection and processing summary.**

| Collected Specimen | Quantity | Biobanked Specimen Type | Number of Aliquots | Aliquot Volume | Processing |
|---|---|---|---|---|---|
| Nasopharyngeal or nasal swab* | 1 | Nasal Cells | NA | 1 | Processed locally within 1 hour; frozen to -90°C to -65°C; batch shipped on dry ice |
| Blood in serum separator tube | 2 x 8.5 ml | Serum | 10 | 1 ml | |
| Blood in cell preparation tube (CPT) ** | 4 x 8 ml | Peripheral Blood Mononuclear Cells (PBMCs) | 8 x PBMCs (target cell count minimum 5 million cells/mL) | 1 ml | Centrifuged locally, refrigerated, shipped at refrigerated temperature on day of collection |
| Blood in sodium citrate tube** | 2 x 2.7 ml | Plasma | 2 | 1 ml | Processed locally within 1 hour; frozen to -90°C to -65°C; batch shipped on dry ice |
| Blood in ethylenediaminetetraacetic acid (EDTA) tube** | 1 x 10 ml | Plasma White blood cells*** | 4 1 | 1 ml | Refrigerated, shipped at refrigerated temperature on day of collection |
| Blood in PAXgene-RNA tube | 1 x 2.5 ml | Whole blood | 1 | 2.5 ml | |
| Urine in no additive tube | 1 x 10 ml | Urine supernatant | 8 | 1 ml | |
| Saliva in Oragene OGR-600*** | 1 x 2 ml | Saliva | 1 | 2 ml | |
| Stool | 1 x 25 ml | Stool | 1 | 25 ml | Directly sent by participant |

* As of June 29, 2022, nasopharyngeal swab changed to nasal swab

** As of protocol v6.0, added sodium citrate and EDTA tubes and CPT reduced to 2 cell preparation tubes.

*** Not collected in those who decline genetic testing

Individuals who are pregnant, 3-months postpartum or breastfeeding are ineligible for some of the assessments.

Tier 2 and 3 assessments include additional surveys, blood tests, clinical examinations, imaging, and procedures. Specialized blood tests are run by a central laboratory (ARUP Laboratories, Salt Lake City, Utah) for consistency. Imaging is acquired via pragmatic standard clinical protocols to maximize testing availability across sites. Several of the imaging studies are overseen by reading centers charged with protocol development, quality assurance/control, and certifying performance sites, in addition to centralized review of a portion of studies. DICOM images are uploaded and shared using specialized cloud-based image storage software provided by Ambra Health. Clinical examinations and procedures are performed by clinically certified personnel at each site following standard clinical protocols.

**Data collection and management.** Study data are collected by sites and entered into a centralized REDCap (Research Electronic Data Capture) database hosted by the DRC in a Federal Information Security Modernization Act moderate environment [43]. REDCap includes data validation and audit capabilities [44, 45]. Protected health information in the central REDCap database is limited to zip code and birthdate. Automated queries are generated by the DRC for missing or implausible data and sent to sites for near real-time correction. Monthly study monitoring reports are provided to sites to optimize fidelity to protocol. Periodic audits are conducted independent of the site investigators and sponsor.

## Outcomes

The primary endpoints of this study are the presence of composite incident or prevalent PASC symptoms and progression of PASC; since there is not yet an agreed-upon definition of PASC, a working definition will be developed as part of the study (see statistical analysis). Secondary endpoints include recovery trajectories from SARS-CoV-2 infection, documentation of organ injury, and incident clinical diagnoses.

## Statistical analysis

Point prevalence, defined as the proportion of participants reporting a symptom at a given follow-up time point among those remaining in RECOVER, will be calculated for participants with and without infection separately. Odds ratios (ORs) adjusted for demographic factors will be reported. Machine learning approaches will be used to select combinations of symptoms among the 40+ included in the symptom survey (i.e., variable selection) that differentiate participants with and without a history of infection [46]. Prevalent symptoms and select severity scores will be used as input to the model. Balancing weights will be applied to account for any differences between infected and uninfected participants [47]. Analyses will be iteratively refined as new data modalities (e.g. laboratory, radiology and other Tier 2/3 tests) and additional longitudinal assessments become available. Within PASC positive individuals, consensus clustering will be applied to identify PASC subgroups [48].

Logistic regression analyses will be conducted to investigate associations between clinical factors, demographics, and SDoH, and the cumulative incidence of PASC among participants with infection. Multinomial regression models will also be used to investigate specific associations between these risk factors and PASC subgroups. A Cox proportional hazards regression will be fitted to model the hazard of developing PASC given the risk factors. A Fine-Gray model for the sub-distributional hazard of developing each PASC subgroups, accounting for the competing and semi-competing risks of each sub-phenotype as well as study dropout as a censoring event, will be fit among infected participants to estimate the association between risk factors and the hazard of each PASC subgroups [49]. Additional strategies that account

for time-varying covariates (e.g., vaccination status, pharmaceutical or clinical interventions) will be considered [50].

A Cox proportional hazard model will model PASC resolution, defined based on longitudinal assessments, to evaluate associations with baseline factors, including markers of illness severity during the acute phase of infection. Point estimates and 95% Wald confidence intervals will be estimated for each risk factor and a large-sample score test will be conducted to test against the null hypothesis that the hazard of resolution is independently associated with each risk factor.

### Observational study monitoring board

The RECOVER OSMB appointed by the NIH provides data and safety oversight, meeting at least twice annually. The purpose of the OSMB is to assure independent review of unreasonable risk exposure because of study participation, monitor study progress and integrity, and advise on significant protocol modifications. The OSMB is composed of experts in longitudinal studies, manifestations of COVID-19, biostatistics and bioethics, and patient/caregiver representatives. As RECOVER-Adult does not involve any interventions, early stopping rules for efficacy or futility are not indicated.

### Major changes to the protocol

A planned flexible study design allows modifications to PASC case definition, tiered phenotyping assessments, comparator groups, and statistical plan after study initiation to optimize public health impact without undermining validity and integrity of study findings. Table 4 lists key modifications to the protocol to date.

### Data sharing and dissemination

Scientific data will be de-identified and shared pursuant to the NIH policy for Data Management and Sharing policy [51]. The RECOVER Ancillary Study Oversight Committee and Biospecimen Access Committee govern access to data and biospecimens for ancillary studies. Study results will be disseminated via scientific publication, presentation at national meetings, public-facing webinars, community briefings, RECOVER newsletters, social media, and other means of communication to both scientific and lay audiences. Additionally, results of all tests performed in CLIA-certified laboratories or read by clinically-certified personnel are returned to study participants [52]. Tests performed in research laboratories that are not CLIA-certified are not returned to participants, following federal regulations.

### Ethics

The study was approved by the NYU Grossman School of Medicine Institutional Review Board (IRB), which serves as the single IRB for the majority of the study sites. A few pre-existing consortia use their own IRBs through an exemption granted by the NIH. All participants provide signed, informed consent to participate in the main protocol. Participants are reconsented if there are major changes to the study design or to anticipated risks. For high-risk Tier 3 procedures, a separate procedure-specific consent is obtained prior to the procedure.

### Discussion

The overall goal of RECOVER is to rapidly improve understanding of, and ability to predict, treat, and prevent PASC. RECOVER-Adult's large sample size and breadth of representation across geographic region, age, sex, race/ethnicity and other SDoH and pregnancy status are

Table 4. Selected, key protocol modifications since initial approval.

| Protocol version | Date of approval | Major changes | Participants enrolled at time of modification |
|---|---|---|---|
| 1.0 | 9/16/2021 | Original release | 0 |
| 1.1 | 10/08/2021 | Additions to comorbidity form to align with NeuroCOVID Databank | 0 |
| | | Orthostatic test changed to active stand test | |
| 2.0 | 10/18/2021 | Coagulation panel, urinalysis, SARS-CoV-2 PCR (uninfected) added to Tier 1 | 0 |
| | | Mini International Neuropsychiatric Interview, adrenocorticotropic hormone, morning cortisol, hepatitis B and C, renal ultrasound added to Tier 2 | |
| | | Electrocardiogram, D-dimer, troponin, NT-pro BNP moved from Tier 2 to Tier 1 | |
| | | Tilt table testing, cardiovagal innervation testing, catecholamine testing moved from Tier 2 to Tier 3 | |
| | | Ear-nose-throat exam, lung plethysmography, dual energy chest CT moved from Tier 3 to Tier 2 | |
| | | Parathyroid hormone, gamma-glutamyl transferase, anti-phospholipid antibody, CT pulmonary angiography, ventilation/perfusion scan, MRI spine, abdominal CT removed from protocol | |
| 3.0 | 12/01/2021 | Clarified inclusion criteria to include patients who have positive SARS-Cov-2 infection-specific antibody testing | 30 |
| | | Added incarceration as an exclusion criterion. | |
| | | Added an opt-in for future genetic testing | |
| | | Removed procalcitonin and moved electrocardiogram from Tier 1 back to Tier 2 | |
| | | Added Anti-Mullerian hormone. | |
| 4.0 | 01/27/2022 | Added one-time off schedule visit during on-study infection | 399 |
| | | Removed option to return research lab results | |
| 5.0 | 03/22/2022 | Updated the recruitment window from 24 to 36 months since first infection | 1,474 |
| | | Added that uninfected pregnant individuals begin study schedule on delivery date | |
| | | Added that women <3 months postpartum can not have any tests that pregnant women can not have | |
| | | Specified methylmalonic acid to be drawn with serum B12 | |
| 6.0 | 8/11/2022 | Revised the earliest date of possible infection from March to January, 2020 | 7,698 |
| | | Removed lung plethysmography | |
| | | Removed 15% cap on self-referral participants | |
| | | Changed study schedule to start at time of acute reinfection (enrollment) for previously infected participants enrolling during an acute infection | |
| | | Reduced the number of cell preparation tubes for collection and replaced with sodium citrate and EDTA tubes for plasma | |
| | | Moved anti-nuclear antibody, anti-cyclic citrullinated peptide antibodies, rheumatoid factor, Epstein Barr virus testing to Tier 2 | |
| 7.0 | 12/15/2022 | Added collection of tears as a biospecimen | 11,602 |
| | | Added section on the mobile health platform | |
| 8.0 | Pending | Reduced target N of acute infected enrollees from 7,800 to 5,000 given higher than expected PASC rates | |

expected to produce broadly applicable and actionable results and support numerous sub-group analyses. Additionally, nested case-control studies will occur among participants with certain PASC phenotypes who have triggered assessments. RECOVER-Adult includes numerous strengths. Participants include many uninfected and asymptomatic infected individuals for comparison that is often missing from other large studies [11, 53–56]. All participants are followed prospectively from time of enrollment, allowing longitudinal analyses of disease trajectory. By contrast, most studies to date have been single time point assessments or serial cross-sectional studies [3–9, 57]. Acute participants enrolled at the time of first infection will provide a prospective estimate of PASC rates that is less biased than the most studies, which have enrolled subjects after PASC status is known [3–9]. The strong focus on patient-reported

symptom outcomes allows capture of a broader range of sequelae than studies relying on electronic health records or claims data [53, 58–62]. The adaptive nature of the protocol allows for rapid responsiveness to new discoveries and changes in the nature of the pandemic. The extensive biospecimen collection and clinical, laboratory, and radiology assessments will generate a wealth of deep phenotyping data that can be used for pathophysiologic analyses. Finally, multi-omics analyses are proposed and have potential to provide molecular mechanistic insights into the pathophysiology of PASC.

RECOVER-Adult is also unique in the extent to which patients experiencing PASC and representatives from patient advocacy communities contributed to the protocol's development and ongoing operations. For example, the PASC symptom survey was drawn in part from lists of symptoms generated by members of the patient community [2], allowing measurement of symptoms overlooked by other studies, including post-exertional malaise and menstrual cycle changes. At the urging of patient representatives, participants with a clinical diagnosis of COVID were included even without a positive test history, to permit inclusion of individuals affected in the earliest stages of the pandemic when testing was not widely available. Among the many other significant design aspects credited to input from patient representatives are: wording of the consent document, including clinical assessments specific to dysautonomia, sharing clinically certified lab results with participants, ensuring accommodations for participants with myalgic encephalomyelitis/chronic fatigue syndrome, and selection of SDoH instruments.

In summary, RECOVER-Adult is a large, national, longitudinal, retrospective and prospective cohort that will answer key questions about the epidemiology and pathophysiology of PASC. Results will support clinical trial development by defining PASC and sub-phenotypes, natural history, risk factors, biomarkers, and mechanistic pathways for potential therapeutic targets. Results of this study will also inform public health efforts, prevention, and clinical care.

## Supporting information

**S1 Fig. Protocol development timeline.**
(DOCX)

**S2 Fig. RECOVER consortium oversight structure.**
(DOCX)

**S1 Table. Hubs and enrolling sites.**
(DOCX)

**S2 Table. Survey topics as of protocol version 7.0.**
(DOCX)

**S3 Table. Tier 1 assessments.**
(DOCX)

**S4 Table. Tier 2 assessments.**
(DOCX)

**S5 Table. Tier 3 assessments.**
(DOCX)

**S6 Table. Writing committee.**
(DOCX)

**S7 Table. RECOVER-Adult consortium members.**
(DOCX)

**S8 Table. RECOVER-Adult committees and task forces.**
(DOCX)

**S1 Checklist.**
(DOCX)

**S1 File.**
(PDF)

**S2 File.**
(PDF)

# Acknowledgments

We would like to thank the National Community Engagement Group (NCEG), all patient, caregiver and community representatives, and all the participants enrolled in the RECOVER initiative.

# Author Contributions

**Conceptualization:** Leora I. Horwitz, Tanayott Thaweethai, Megan L. Fitzgerald, Jason D. Goldman, Rachel Hess, S. L. Hodder, Vanessa L. Jacoby, Michael R. Jordan, Jerry A. Krishnan, Adeyinka O. Laiyemo, Torri D. Metz, Rachel E. Patzer, Nora G. Singer, Lauren E. Stiles, Lisa Aponte-Soto, Ingrid V. Bassett, Andra L. Blomkalns, Hassan Brim, Alexander W. Charney, Lori B. Chibnik, Rebecca G. Clifton, Steven G. Deeks, Kristine M. Erlandson, Julio C. Facelli, Jennifer C. Gander, James R. Heath, Judith A. James, Sarah Jolley, Elizabeth W. Karlson, Stuart D. Katz, J. Daniel Kelly, John P. Kirwan, Kenneth S. Knox, Andre Kumar, Bruce D. Levy, Gailen D. Marshall, Jeffrey N. Martin, Grace A. McComsey, Lucio Miele, Sindhu Mohandas, Girish N. Nadkarni, Janko Z. Nikolich, Igho Ofotokun, Sairam Parthasarathy, Thomas F. Patterson, Priscilla Pemu, Athena Poppas, Davin K. Quinn, Hengameh Raissy, Clifford J. Rosen, Frank C. Sciurba, Sudha Seshadri, Howard D. Sesso, Dimpy P. Shah, Upinder Singh, Larissa J. Teunis, Alan T. N. Tita, P. J. Utz, John C. Wood, Andrea S. Foulkes.

**Data curation:** Tanayott Thaweethai, Shifa Ahmed, Heather A. Algren, Hassan Ashktorab, Marie-Abele C. Bind, James Chan, Alexander W. Charney, Dominic C. Chow, Helen Y. Chu, Sarah E. Donohue, Kristine M. Erlandson, Frank L. Greenway, Jenny E. Han, Matthew K. Hoffman, Stuart D. Katz, J. Daniel Kelly, Andre Kumar, Joyce K. Lee-lannotti, Jennifer K. Logue, Amber Nguyen, Anna Palatnik, Kristy T. S. Palomares, Harrison T. Reeder, Dwight J. Rouse, Caitlin Selvaggi, Howard D. Sesso, Dimpy P. Shah, Zaki A. Sherif, Daniel J. Shinnick, John M. Thorp, Jr., Alan T. N. Tita, Tiffany Walker, Andrea S. Foulkes.

**Formal analysis:** Rachel Hess, Lori B. Chibnik.

**Funding acquisition:** Leora I. Horwitz, Tanayott Thaweethai, Rachel Hess, Jerry A. Krishnan, Adeyinka O. Laiyemo, Torri D. Metz, Lauren E. Stiles, Barbara S. Taylor, Ingrid V. Bassett, Nahid Bhadelia, Christian Bime, Alexander W. Charney, Helen Y. Chu, Steven G. Deeks, Julio C. Facelli, Joey P. Granger, John W. Hafner, James R. Heath, Judith A. James, Stuart D. Katz, J. Daniel Kelly, John P. Kirwan, Kenneth S. Knox, Bruce D. Levy, Vincent C. Marconi, Christopher F. Martin, Jeffrey N. Martin, Grace A. McComsey, Girish N. Nadkarni, Lisa T. Newman, Igho Ofotokun, Priscilla Pemu, Dwight J. Rouse, Frank C. Sciurba, Dimpy P. Shah, Zaki A. Sherif, Alan T. N. Tita, David E. Warren, John C. Wood, Andrea S. Foulkes.

**Investigation:** Leora I. Horwitz, Jason D. Goldman, Rachel Hess, Vanessa L. Jacoby, Jerry A. Krishnan, Adeyinka O. Laiyemo, Torri D. Metz, Rachel E. Patzer, Nora G. Singer, Barbara S. Taylor, Heather A. Algren, Lisa Aponte-Soto, Ingrid V. Bassett, Nahid Bhadelia, Christian Bime, Hassan Brim, Mario Castro, Alexander W. Charney, Peter Chen, Helen Y. Chu, Maged M. Costantine, Sushma K. Cribbs, Steven G. Deeks, John D. Dickinson, Matthew S. Durstenfeld, Valerie J. Flaherman, Vivian Fonseca, Jennifer C. Gander, Minjoung Go, Frank L. Greenway, John W. Hafner, Jenny E. Han, Michelle S. Harkins, James R. Heath, On Ho, Carol R. Horowitz, Harvey Hsu, Priscilla Y. Hsue, Prasanna Jagannathan, Janice John, Joy J. Juskowich, J. Daniel Kelly, Arthur Y. Kim, John P. Kirwan, Joyce K. Lee-lannotti, Bruce D. Levy, Janet Y. Lin, Carlos A. Luciano, Gail Mallett, Vincent C. Marconi, Gailen D. Marshall, Jeffrey N. Martin, Grace A. McComsey, Hector Mendez-Figueroa, Sindhu Mohandas, Janet M. Mullington, Girish N. Nadkarni, Igho Ofotokun, Anna Palatnik, Tanyalak Parimon, Samuel Parry, Sairam Parthasarathy, Thomas F. Patterson, Ann Pearman, Michael J. Peluso, Priscilla Pemu, Beth A. Plunkett, Kristen Pogreba-Brown, John G. Quigley, Davin K. Quinn, Hengameh Raissy, Candida J. Rebello, Uma M. Reddy, Franz P. Rischard, Nadine G. Rouphael, Jorge L. Santana, Frank C. Sciurba, Howard D. Sesso, Dimpy P. Shah, Zaki A. Sherif, Hyagriv N. Simhan, Upinder Singh, Amber Sowles, Vignesh Subbian, John M. Thorp, Jr., Amberly Ticotsky, Alan T. N. Tita, Robin Tragus, Alfredo E. Urdaneta, P. J. Utz, Timothy M. VanWagoner, Honorine D. Ward, Sidney W. Whiteheart, John C. Wood, Lynn M. Yee, Sokratis N. Zisis.

**Methodology:** Leora I. Horwitz, Tanayott Thaweethai, Shari B. Brosnahan, Megan L. Fitzgerald, Jason D. Goldman, Rachel Hess, Michael R. Jordan, Jerry A. Krishnan, Torri D. Metz, Lauren E. Stiles, Barbara S. Taylor, Shifa Ahmed, Ingrid V. Bassett, Brahmchetna Bedi, Nahid Bhadelia, Christian Bime, Marie-Abele C. Bind, James Chan, Alexander W. Charney, Lori B. Chibnik, Kristine M. Erlandson, Valerie J. Flaherman, Jennifer C. Gander, On Ho, Carol R. Horowitz, Sarah Jolley, S. E. Judd, Elizabeth W. Karlson, Stuart D. Katz, J. Daniel Kelly, John P. Kirwan, Joyce K. Lee-lannotti, Bruce D. Levy, Gabrielle Maranga, Gailen D. Marshall, Jeffrey N. Martin, Grace A. McComsey, Lucio Miele, Sindhu Mohandas, Janet M. Mullington, Erica R. Nahin, Amber Nguyen, Igho Ofotokun, Sairam Parthasarathy, Michael J. Peluso, Athena Poppas, J. Zachary Porterfield, Davin K. Quinn, Hengameh Raissy, Harrison T. Reeder, Franz P. Rischard, Johana M. Rosas, Christina Saint Jean, Caitlin Selvaggi, Sudha Seshadri, Howard D. Sesso, Dimpy P. Shah, Eyal Shemesh, Zaki A. Sherif, Daniel J. Shinnick, Mehul S. Suthar, Alan T. N. Tita, P. J. Utz, Suzanne D. Vernon, David E. Warren, Jordan C. Weyer, Juan P. Wisnivesky, Andrea S. Foulkes.

**Project administration:** Leora I. Horwitz, Tanayott Thaweethai, Shari B. Brosnahan, Mine S. Cicek, Rachel Hess, S. L. Hodder, Jerry A. Krishnan, Adeyinka O. Laiyemo, Barbara S. Taylor, Shifa Ahmed, Heather A. Algren, Brahmchetna Bedi, Nahid Bhadelia, Marie-Abele C. Bind, Lora J. Black, Hassan Brim, James Chan, Alexander W. Charney, Benjamin K. Chen, Li Qing Chen, Rebecca G. Clifton, Shelby Collins, John D. Dickinson, Sarah E. Donohue, Ivette F. Emery, Kristine M. Erlandson, Julio C. Facelli, Rachael Farah-Abraham, Melinda S. Fischer, Judes Fleurimont, Jennifer C. Gander, Minjoung Go, Joey P. Granger, Frank L. Greenway, Jenny E. Han, Kristine S. P. Hauser, James R. Heath, Carla R. Hernandez, On Ho, Janice John, Diane G. Kanjilal, Stuart D. Katz, J. Daniel Kelly, John P. Kirwan, Michelle F. Lamendola-Essel, Joyce K. Lee-lannotti, R. Craig Lefebvre, Janet Y. Lin, Jennifer K. Logue, Gabrielle Maranga, Jai G. Marathe, Vincent C. Marconi, Christopher F. Martin, Jeffrey N. Martin, Janet M. Mullington, Girish N. Nadkarni, Robert B. Neuman, Lisa T. Newman, Amber Nguyen, Janko Z. Nikolich, Igho Ofotokun, Anna Palatnik, Sairam Parthasarathy, Ann Pearman, Michael J. Peluso, J. Zachary Porterfield, Hengameh Raissy,

Harrison T. Reeder, Nadine G. Rouphael, Adam M. Ruff, Christina Saint Jean, Shannon M. Schlater, Caitlin Selvaggi, Howard D. Sesso, Eyal Shemesh, Daniel J. Shinnick, Hyagriv N. Simhan, Upinder Singh, Amber Sowles, Vignesh Subbian, Larissa J. Teunis, John M. Thorp, Jr., Alan T. N. Tita, Robin Tragus, P. J. Utz, Suzanne D. Vernon, Crystal Vidal, Honorine D. Ward, David E. Warren, Steven J. Weiner, Jordan C. Weyer, Natasha J. Williams, John C. Wood, Lynn M. Yee, Andrea S. Foulkes.

**Resources:** Leora I. Horwitz, Tanayott Thaweethai, Mine S. Cicek, Jerry A. Krishnan, Shifa Ahmed, Heather A. Algren, Christian Bime, Marie-Abele C. Bind, Andra L. Blomkalns, James Chan, Alexander W. Charney, Steven G. Deeks, Ivette F. Emery, Julio C. Facelli, Jennifer C. Gander, Frank L. Greenway, Jenny E. Han, Kristine S. P. Hauser, James R. Heath, Carla R. Hernandez, Stuart D. Katz, J. Daniel Kelly, John P. Kirwan, Michelle F. Lamendola-Essel, R. Craig Lefebvre, Gabrielle Maranga, Vincent C. Marconi, Sindhu Mohandas, Christian Mouchati, Girish N. Nadkarni, Amber Nguyen, Janko Z. Nikolich, Princess U. Ogbogu, Sairam Parthasarathy, Thomas F. Patterson, J. Zachary Porterfield, John G. Quigley, Harrison T. Reeder, Caitlin Selvaggi, Zaki A. Sherif, Daniel J. Shinnick, Vignesh Subbian, Alan T. N. Tita, P. J. Utz, Honorine D. Ward, Lynn M. Yee, Andrea S. Foulkes.

**Software:** Leora I. Horwitz, Tanayott Thaweethai, Shifa Ahmed, Marie-Abele C. Bind, James Chan, Amber Nguyen, Harrison T. Reeder, Caitlin Selvaggi, Daniel J. Shinnick, Andrea S. Foulkes.

**Supervision:** Leora I. Horwitz, Tanayott Thaweethai, Rachel Hess, S. L. Hodder, Vanessa L. Jacoby, Jerry A. Krishnan, Rachel E. Patzer, Barbara S. Taylor, Heather A. Algren, Ingrid V. Bassett, Brahmchetna Bedi, Nahid Bhadelia, Marie-Abele C. Bind, Hassan Brim, Mario Castro, James Chan, Alexander W. Charney, Peter Chen, Helen Y. Chu, Shelby Collins, Maged M. Costantine, Steven G. Deeks, Ivette F. Emery, Kristine M. Erlandson, Julio C. Facelli, Rachael Farah-Abraham, Vivian Fonseca, Jennifer C. Gander, Kristine S. P. Hauser, James R. Heath, Carla R. Hernandez, On Ho, Matthew K. Hoffman, Prasanna Jagannathan, Judith A. James, Diane G. Kanjilal, Stuart D. Katz, J. Daniel Kelly, John P. Kirwan, Kenneth S. Knox, Michelle F. Lamendola-Essel, Joyce K. Lee-Iannotti, Michele T. Longo, Carlos A. Luciano, Gabrielle Maranga, Vincent C. Marconi, Gailen D. Marshall, Christopher F. Martin, Jeffrey N. Martin, Grace A. McComsey, Hector Mendez-Figueroa, Janet M. Mullington, Robert B. Neuman, Lisa T. Newman, Janko Z. Nikolich, Sairam Parthasarathy, Thomas F. Patterson, Beth A. Plunkett, Athena Poppas, Davin K. Quinn, Harrison T. Reeder, Dwight J. Rouse, Christina Saint Jean, Frank C. Sciurba, Caitlin Selvaggi, Hyagriv N. Simhan, Upinder Singh, Vignesh Subbian, Larissa J. Teunis, Alan T. N. Tita, Katherine R. Tuttle, P. J. Utz, Crystal Vidal, Tiffany Walker, Honorine D. Ward, Jordan C. Weyer, John C. Wood, Lynn M. Yee, Andrea S. Foulkes.

**Validation:** Leora I. Horwitz, David Chestek, J. Daniel Kelly, Jorge L. Santana, Howard D. Sesso, Tiffany Walker.

**Visualization:** Leora I. Horwitz, Nora G. Singer.

**Writing – original draft:** Leora I. Horwitz, Tanayott Thaweethai, Shari B. Brosnahan, Mine S. Cicek, Megan L. Fitzgerald, Rachel Hess, S. L. Hodder, Michael R. Jordan, Jerry A. Krishnan, Adeyinka O. Laiyemo, Rachel E. Patzer, Nora G. Singer, Lori B. Chibnik, Igho Ofotokun, Hengameh Raissy, Andrea S. Foulkes.

**Writing – review & editing:** Tanayott Thaweethai, Shari B. Brosnahan, Mine S. Cicek, Megan L. Fitzgerald, Jason D. Goldman, Rachel Hess, S. L. Hodder, Vanessa L. Jacoby, Michael R. Jordan, Jerry A. Krishnan, Adeyinka O. Laiyemo, Torri D. Metz, Lauren Nichols, Rachel E.

Patzer, Anisha Sekar, Nora G. Singer, Lauren E. Stiles, Barbara S. Taylor, Shifa Ahmed, Heather A. Algren, Khamal Anglin, Lisa Aponte-Soto, Hassan Ashktorab, Ingrid V. Bassett, Christian Bime, Marie-Abele C. Bind, Lora J. Black, Andra L. Blomkalns, Hassan Brim, James Chan, Peter Chen, David Chestek, Lori B. Chibnik, Dominic C. Chow, Helen Y. Chu, Rebecca G. Clifton, Maged M. Costantine, Sushma K. Cribbs, Steven G. Deeks, Sarah E. Donohue, Matthew S. Durstenfeld, Ivette F. Emery, Kristine M. Erlandson, Julio C. Facelli, Aloke V. Finn, Vivian Fonseca, Emily J. Gallagher, Jennifer C. Gander, Maria Laura Gennaro, Kelly S. Gibson, Steven N. Goodman, Frank L. Greenway, John W. Hafner, On Ho, Susan E. Hoover, Carol R. Horowitz, Harvey Hsu, Brenna L. Hughes, Prasanna Jagan-nathan, Judith A. James, Janice John, Sarah Jolley, S. E. Judd, Joy J. Juskowich, Diane G. Kanjilal, Elizabeth W. Karlson, Stuart D. Katz, J. Daniel Kelly, Sara W. Kelly, Arthur Y. Kim, John P. Kirwan, Kenneth S. Knox, Margaret Lanca, Joyce K. Lee-lannotti, R. Craig Lefebvre, Bruce D. Levy, Janet Y. Lin, Brian P. Logarbo, Jr., Jennifer K. Logue, Carlos A. Luciano, Karen Lutrick, Shahdi K. Malakooti, Jai G. Marathe, Vincent C. Marconi, Jeffrey N. Martin, Heidi T. May, Grace A. McComsey, Dylan McDonald, Lucio Miele, Murray A. Mittleman, Sindhu Mohandas, Janet M. Mullington, Girish N. Nadkarni, Erica R. Nahin, Amber Nguyen, Janko Z. Nikolich, Igho Ofotokun, Princess U. Ogbogu, Samuel Parry, Sairam Parthasarathy, Thomas F. Patterson, Ann Pearman, Michael J. Peluso, Priscilla Pemu, Christian M. Pettker, Beth A. Plunkett, Kristen Pogreba-Brown, J. Zachary Porter-field, John G. Quigley, Hengameh Raissy, Candida J. Rebello, Uma M. Reddy, Rebecca Reece, Harrison T. Reeder, Franz P. Rischard, Johana M. Rosas, Clifford J. Rosen, Nadine G. Rouphael, Dwight J. Rouse, Adam M. Ruff, Grecio J. Sandoval, Frank C. Sciurba, Caitlin Selvaggi, Sudha Seshadri, Howard D. Sesso, Dimpy P. Shah, Daniel J. Shinnick, Hyagriv N. Simhan, Vignesh Subbian, Jun Sun, John M. Thorp, Jr., Alan T. N. Tita, Katherine R. Tuttle, Timothy M. VanWagoner, Andrew Vasey, Suzanne D. Vernon, Tiffany Walker, Honorine D. Ward, David E. Warren, Ryan M. Weeks, Steven J. Weiner, Jordan C. Weyer, Jennifer L. Wheeler, Zanthia Wiley, Natasha J. Williams, Juan P. Wisnivesky, John C. Wood, Lynn M. Yee, Natalie M. Young, Sokratis N. Zisis, Andrea S. Foulkes.

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
