## [Decision Letter · Decision Letter 0]

2 May 2023

PONE-D-23-08595Researching COVID to enhance recovery (RECOVER) adult study protocol: Rationale, objectives, and designPLOS ONE

Dear Dr. Horwitz,

Thank you for submitting your manuscript to PLOS ONE. After careful consideration, we feel that it has merit but does not fully meet PLOS ONE’s publication criteria as it currently stands. Therefore, we invite you to submit a revised version of the manuscript that addresses the points raised during the review process.

We look forward to receiving your revised manuscript.

Kind regards,

Luis Felipe Reyes, M.D., Ph.D., MSc.

Academic Editor

PLOS ONE

Journal Requirements:

"We would like to thank the National Community Engagement Group (NCEG), all patient, caregiver and community representatives, and all the participants enrolled in the RECOVER initiative.

Membership of the RECOVER Initiative is provided in a supplemental appendix file."

"National Institutes of Health (NIH) Agreement OTA OT2HL161847 (SDK, LIH), OT2HL161841 (ASF), OT2HL156812 (LTN). https://www.nih.gov/

The funders did not and will not have a role in study design, data collection and analysis, decision to publish, or preparation of the manuscript."

Reviewers' comments:

Reviewer's Responses to Questions

**Comments to the Author**

1. Does the manuscript provide a valid rationale for the proposed study, with clearly identified and justified research questions?

Reviewer #1: Yes

Reviewer #2: Yes

2. Is the protocol technically sound and planned in a manner that will lead to a meaningful outcome and allow testing the stated hypotheses?

Reviewer #1: Yes

Reviewer #2: Yes

3. Is the methodology feasible and described in sufficient detail to allow the work to be replicable?

Reviewer #1: Yes

Reviewer #2: Yes

4. Have the authors described where all data underlying the findings will be made available when the study is complete?

Reviewer #1: Yes

Reviewer #2: Yes

5. Is the manuscript presented in an intelligible fashion and written in standard English?

Reviewer #1: Yes

Reviewer #2: Yes

6. Review Comments to the Author

You may also provide optional suggestions and comments to authors that they might find helpful in planning their study.

Reviewer #1: Sample size: The primary aims of this study are more focused on characterizing the prevalence, symptoms, and other factors related to PASC. Therefore, the sample size should not be based solely on comparisons between infected and non-infected individuals. Additionally, the effect sizes used in the calculation were not clinically justified. It is unclear how the targeted sample sizes were determined for subgroups in Table 2.

The purpose and technique details of the machine learning model are not clearly described.

It is unclear how the PASC subgroups will be analyzed.

A longitudinal model may be considered for analyzing repeated measures.

Reviewer #2: It is a well written protocol. It has ambitious but achievable objectives. The methodology is adequate. The large number of participating centers ensures the achievement of the sample size. The most important limitation is the lack of a standardized definition of long COVID-19. However, with the findings of this study a better characterization could be achieved.

7. PLOS authors have the option to publish the peer review history of their article (what does this mean?). If published, this will include your full peer review and any attached files.

Reviewer #1: No

Reviewer #2: No

---

## [Author Response · Author response to Decision Letter 0]

11 May 2023

please see Response to Reviewers document, attached

---

## [Editor Report · Decision Letter 1]

15 May 2023

Researching COVID to enhance recovery (RECOVER) adult study protocol: Rationale, objectives, and design

PONE-D-23-08595R1

Dear Dr. Horwitz,

We’re pleased to inform you that your manuscript has been judged scientifically suitable for publication and will be formally accepted for publication once it meets all outstanding technical requirements.

Kind regards,

Luis Felipe Reyes, M.D., Ph.D., MSc.

Academic Editor

PLOS ONE
---

## [Editor Report · Acceptance letter]

7 Jun 2023

PONE-D-23-08595R1 

Researching COVID to enhance recovery (RECOVER) adult study protocol: Rationale, objectives, and design 

Dear Dr. Horwitz:

I'm pleased to inform you that your manuscript has been deemed suitable for publication in PLOS ONE. Congratulations! Your manuscript is now with our production department. 

Kind regards, 

on behalf of

Dr. Luis Felipe Reyes 

Academic Editor

PLOS ONE